# Two Cases of Severe Complications Due to an Esophageal Fish Bone Foreign Body

**DOI:** 10.3390/medicina59091504

**Published:** 2023-08-22

**Authors:** Ji-Hee Han, Ra-Ri Cha, Ji-Yoon Kwak, Hankyu Jeon, Sang-Soo Lee, Jae Jun Jung, Jin Kyu Cho, Hyun Jin Kim

**Affiliations:** 1Department of Internal Medicine, Gyeongsang National University Changwon Hospital, Gyeongsang National University School of Medicine, Jinju 52828, Republic of Korea; hanjisky@naver.com (J.-H.H.); jiuni_01@naver.com (J.-Y.K.); polaris739@naver.com (H.J.); 3939lee@naver.com (S.-S.L.); imdrkim@naver.com (H.J.K.); 2Department of Thoracic and Cardiovascular Surgery, Gyeongsang National University Changwon Hospital, Gyeongsang National University School of Medicine, Jinju 52828, Republic of Korea; thoracoscope@naver.com; 3Department of General Surgery, Gyeongsang National University Hospital, Gyeongsang National University School of Medicine, Jinju 52828, Republic of Korea; rett0322@naver.com

**Keywords:** foreign body, esophageal perforation, endoscopic submucosal dissection, case report, percutaneous drainage, esophageal abscess

## Abstract

Cases of foreign body ingestion are encountered relatively often in clinical settings; however, serious complications are rare. In such cases, mediastinal abscess due to esophageal perforation can become a life-threatening complication. We encountered two cases of severe complications due to an esophageal fish bone foreign body. The first case was a 40-year-old male with an intramural esophageal abscess due to a fish bone after eating fish five days before visiting the hospital. The patient underwent surgical treatment, but the esophageal abscess did not improve; so, the abscess was drained through endoscopic mucosal dissection, and the abscess improved. In the second case, a 64-year-old male, who had eaten fish three days before visiting the hospital, had esophageal perforation by a fish bone, and abscess formation in the mediastinum and the lesser sac in the abdominal cavity were observed. Although surgical treatment was performed, the intra-abdominal abscess formation was not controlled; so, percutaneous drainage (PCD) was inserted, and the abscess improved. Both patients were discharged without any complications. Here, we report two cases that were improved through surgical treatments and additional treatments such as endoscopic dissection and PCD.

## 1. Introduction

The ingestion of a foreign body (FB), including food bolus impaction, is often encountered in clinical practice [1]. Fish bones are a very common cause of FB diseases in adults in Asia [2]. Esophageal fish bone FB diseases have a wide spectrum of clinical manifestations, from minor diseases that can resolve spontaneously to severe fatal diseases that can lead to death. Although rare, these diseases can lead to severe complications, including stricture formation, esophageal perforation, tracheoesophageal fistula, and aortoesophageal fistula [3]. A significant delay in diagnosis can result in fatal complications [4,5]. Various treatments, such as flexible endoscopic removal or surgical procedure have been discussed, but no standard of care has yet been agreed upon [6,7]. Since this clinical condition is so variable, the approach for patients should begin with a complete and detailed clinical history and a physical evaluation to confirm the patient’s symptoms.

In this paper, we report on the occurrence of esophageal perforation by a fish bone FB and its severe complications, and we introduce a treatment for each case.

## 2. Case Presentation

### 2.1. Case 1

This case is an intramural esophageal abscess in a patient with an esophageal foreign body. A 40-year-old male presented to the emergency room with a five-day history of a sore throat and fever. The patient had no underlying disease and had fish soup six days prior. Starting the next day, the patient had throat discomfort but waited for the symptoms to progress. The pain gradually worsened, and the patient came to the emergency room.

Laboratory tests showed leukocytosis (WBC 17,350 μL, normal range: 4500–11,000 μL) and elevated C-reactive protein (CRP, 120.7 mg/L, normal range: 0–5 mg/L). A neck computed tomography (CT) scan showed complicated esophagitis with a 1.2 cm para esophageal abscess formation (Figure 1(A-1,A-2)).

The patient underwent an emergency surgical operation (VATS Rt mediastinotomy and Lt exploration). The entire upper thoracic esophagus was dissected, but there was no abscess formation, and the paratracheal area was also dissected. In this area, the lymph node was enlarged due to inflammation.

An additional neck CT scan was performed the next day. Although the patient underwent a surgical operation and broad-spectrum antibiotic therapy, the neck CT scan showed aggravated paratracheal and paraesophageal areas multiloculated with abscess cavities (Figure 1(B-1,B-2)).

An endoscopy showed diffuse swelling with a small opening discharging pus in the upper esophagus at 23 cm of the upper incisor (UI). The swelling of the esophagus and the gastric cardia were caused by submucosal pus accumulation. A dual knife (Olympus, Tokyo, Japan) was used for the periesophageal abscess drainage, and the mucosal bridge between the true and false lumens from the upper to lower esophagus was endoscopically transected approximately 3 cm vertically. Abscess drainage was confirmed (Figure 2). The drained pus was cultured and stained, but the results were negative.

Two weeks later, a neck CT scan showed a marked decrease in the size of the preexisting multiloculated abscess cavities (Figure 1(C-1,C-2)). A follow up upper endoscopy found a mucosal defect; however, the abscess was improved. In addition, esophagography was performed prior to feeding, and there was no leakage. As a result, endoscopic intraluminal drainage was successfully performed, and the patient was discharged without complications. The patient visited the outpatient clinic 4 weeks after discharge, and there were no symptoms suggestive of esophageal stricture, such as difficulty swallowing. Follow up endoscopy performed thereafter also confirmed healed mucosa without stricture.

### 2.2. Case 2

This case is a distal esophageal perforation with descending peritoneal abscess in a patient with an esophageal fish bone foreign body. A 64-year-old male with no underlying diseases was transferred to the hospital for an uncontrolled esophageal perforation with an abscess. Three days prior, the patient had a fish bone impaction in their throat whilst eating grilled fish, and following this, the patient had chest discomfort, abdominal pain, and dyspnea.

Laboratory tests showed a white blood cell count of 21,670 μL (normal range: 4500–11,000 μL) and elevated C-reactive protein (CRP, >300 mg/L, normal range: 0–5 mg/L). A chest CT performed on the day of admission to the emergency room revealed distal esophageal perforation abscess formation and a pneumomediastinum (Figure 3(A-1,A-2)). After starting broad-spectrum antibiotics, the patient underwent an emergency surgical operation (mediastinotomy via minithoracotomy), performed on the lower part of the thoracic esophagus, and a pus culture and massive irrigation were performed. *Streptococcus anginosus*, *Streptococcus mitis*, and *Streptococcus oralis* were cultured in the results of staining and culture performed on pus collected during surgery. However, no bacteria were identified in the blood culture.

On day three of hospitalization, a chest and abdomen CT scan showed distal esophageal perforation with abscess and infectious fluid in the posterior mediastinum and lesser sac. The upper endoscopy showed a small mucosal opening that appeared to be punctured up to the muscle layer at the 40 cm from the UI (Figure 4A). Percutaneous abscess drainage (PCD) insertion was performed for fluid collection at the Lt. pleural space.

One week later, a chest CT scan showed an increased amount of fluid collection in the lesser sac and greater omentum (Figure 3(B-1,B-2)). On the follow up upper endoscopy, continuous drainage of whitish exudate at the perforation site was observed (Figure 4B). PCD insertion was performed for fluid collection at the Lt. aspect of the stomach and peritoneal space.

After two weeks of hospitalization, a chest CT scan showed decreased fluid collection in the lower paraesophageal area and remnant fluid collection in the lesser omentum and pelvic cavity (Figure 3(C-1,C-2)). On the follow up upper endoscopy, the previously observed orifice was not visible (Figure 4C). Esophagography was performed prior to feeding, and there was no leakage.

After three weeks of hospitalization, the follow up chest and abdomen CT scan showed nearly disappeared fluid collection in the lower paraesophageal area and decreased remnant fluid collection in the lesser omentum and pelvic cavity. The patient was discharged on day 28 without any other complications. A final follow up CT scan was performed two weeks after discharge, and no abscess cavity was observed in the chest or abdominal cavity.

## 3. Discussion

FB ingestion and food bolus impaction occur very frequently in clinical practice. Generally, the ingested FB passes spontaneously and does not require further management [2]. However, it has been estimated that 10–20% of FB ingestion cases require medical interventions, and 1% need surgical interventions [8]. FB affecting the upper gastrointestinal tract can lead to severe complications, even fatal outcomes, without proper treatment. Esophageal perforation is a serious life-threatening complication that is increased by the impaction of animal or fish bones [8,9].

Fish bones are a very common cause of FB diseases in adults in Asia, more so than in Western countries [6,7,10,11]. It can be assumed that the difference in FB distribution between Asian and Western countries is due to the important role of dietary habits and cultural background in this condition. Esophageal FB disease caused by fish bones shows various clinical features, ranging from mild diseases that can be cured naturally to severe and fatal diseases. In Korea, an Asian country, fish bones account for the most (46–72.3%) esophageal FBs [12,13,14]. Both cases reported esophageal perforation by a fish bone and the treatment for its complications. Therefore, as described in the cases, pneumomediastinum and abscess due to perforation of the esophagus had already occurred due to an ingested fish bone foreign body, and surgical intervention had to be selected as the first treatment.

As is known, the risk factors for the complications caused by fish bone FBs increase with longer durations of impaction (>24 h), bone type, and longer bone lengths (>3 cm) [14,15,16]. In a recent study, the rate of endoscopic intervention for the treatment of FBs in the esophagus was very high (63–76%), and the rate of surgery and perforation was relatively high when the time from ingestion to therapeutic intervention was delayed [17]. The patients in both cases introduced here visited the hospital 24 h after ingesting the fish bones, and it seems that the complications occurred due to the delayed time after ingestion. In both cases, the size of the fish bone itself could not be confirmed because the patient visited the hospital several days after ingesting the fish bone foreign body.

Above all, early diagnosis and endoscopic removal of esophageal FBs are most important in reducing the potential complications of embedded FBs. Complications such as perforation or abscesses caused by an FB in the esophagus can be improved through treatment methods tailored to the patient’s condition.

The first case was approached with surgical treatment, but the esophageal abscess did not improve; so, the abscess was drained through endoscopic mucosal dissection, and it improved. Some studies have suggested the possibility and feasibility of endoscopic intraluminal drainage in limited cases of phlegmonous esophagitis although not a complication of esophageal perforation [18]. The patient’s disease extent was limited to the cervical esophageal region. Endoscopic intervention rather than additional surgery was planned. The endoscopy showed a mucosal opening resulting from tracks through which fish bones had already passed, and the pus was effectively discharged through mucosal dissection around hole, which improved without additional procedure. A follow up chest CT scan showed well-healed mucosa and resolution of the paraesophageal abscess.

Endoscopic technology has advanced, permitting less invasive procedures without general anesthesia. Therefore, an endoscopic evaluation should be performed to decide whether to drain a paraesophageal abscess. In cases where drainage is necessary, endoscopic intraluminal drainage could be a useful modality for patients.

The second case was also approached with surgical treatment first, but the intra-abdominal abscess formation was not controlled; so, a PCD was inserted. In previously reported cases, there were intra-abdominal abscesses due to a foreign body, but most of these cases were due to perforation of the GI tract, except for the esophagus [19,20]. Although distal esophageal perforation and resulting intra-abdominal abscesses are uncommon, this case is worth reporting because it has clinical significance.

The abscess in the abdominal cavity was discharged with PCD, and then the position of the PCD was adjusted, and eventually the abscess formation was improved afterward. The lower esophageal sphincter and gastroesophageal junction, where the esophagus is physically narrowed, is one of the places that can be affected by a foreign body [21]. In this case, esophageal perforation by an FB occurred in the distal esophagus, and the abscess, accompanied by a pneumomediastinum, spread to the lesser sac in the abdominal cavity. For the treatment of this intra-abdominal abscess pocket, radiological procedures such as PCD were helpful.

Both patients were discharged without any complications. A delayed diagnosis of esophageal perforation by fish bone can lead to severe complications. Therefore, in these cases, surgical treatments were selected first; however, if the surgical treatment does not completely improve the condition, then it can be improved by an additional endoscopic procedure or a method such as PCD.

## 4. Conclusions

Esophageal FBs caused by fish bones are a common disease in Asian countries, including Korea. It is important to prevent complications from occurring through immediate treatment for the esophageal FB. However, active and timely treatment according to the patient’s clinical situation can help improve the prognosis for the esophageal FB with delays and complications. Here, we report two cases that were improved not only through surgical treatments but also additional treatments, such as endoscopic dissection and PCD insertion.

## Figures and Tables

**Figure 1 medicina-59-01504-f001:**
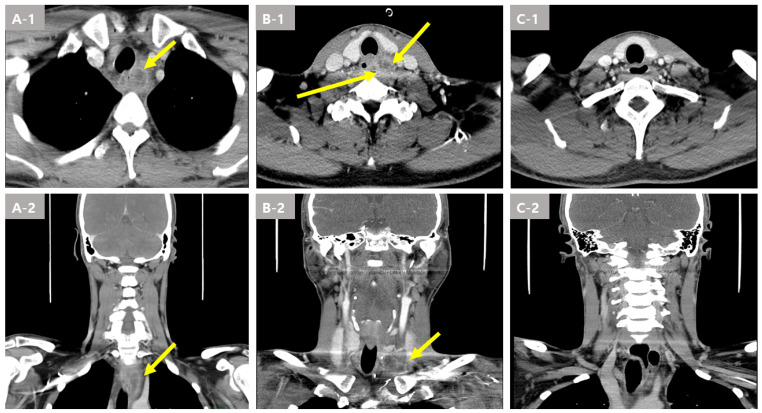
Image findings of the esophageal abscess due to a fish bone foreign body in Case 1. (**A-1**,**A-2**) A complicated esophagitis with paraesophageal abscess formation (arrow) in the esophagus was revealed by a computer tomography (CT) scan at the local hospital. (**B-1**,**B-2**) paratracheal and paraesophageal areas; multiloculated abscess cavities (arrow) in the esophagus were revealed by the CT scan at 2 days. (**C-1**,**C-2**) Markedly decreased size of the preexisting multiloculated abscess cavities in the left visceral space and left sided mediastinum by the CT scan.

**Figure 2 medicina-59-01504-f002:**
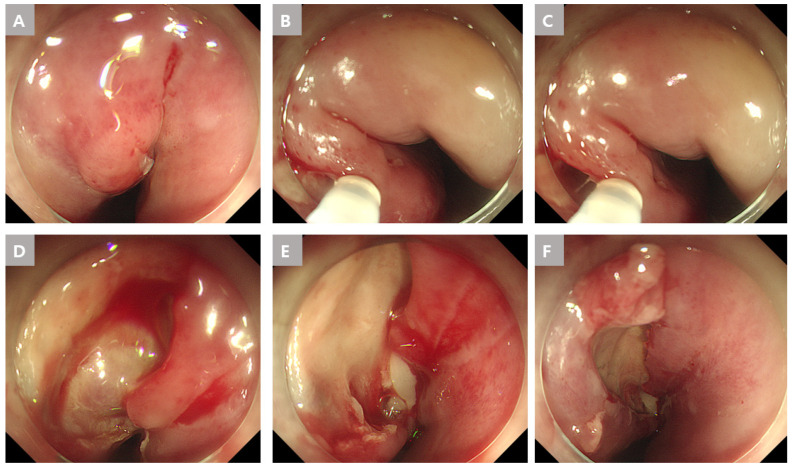
Endoscopic findings of the esophageal abscess due to an esophageal fish bone foreign body in Case 1. (**A**) A mucosal protrusion with a small ulcer was observed with endoscopy. (**B**,**C**) Endoscopic dissection of the mucosa and submucosa, using a dual knife. (**D**–**F**) Entirely exposed submucosal layer and drainage of the abscess.

**Figure 3 medicina-59-01504-f003:**
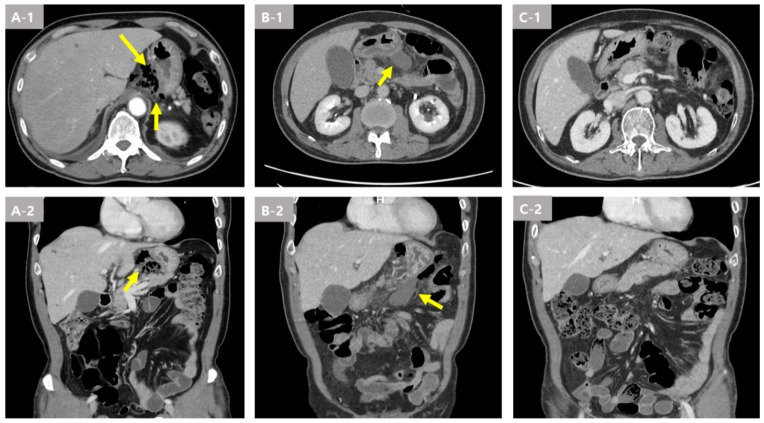
Image findings of the distal esophageal perforation with descending peritoneal abscess due to a fish bone foreign body in Case 2. (**A-1**,**A-2**) A distal esophageal perforation abscess formation and pneumomediastinum was seen by computer tomography (CT) scan. (**B-1**,**B-2**) Increase in the amount of fluid collection in the lesser sac and greater omentum was revealed by the CT scan 1 week later. (**C-1**,**C-2**) Nearly disappeared fluid collection in the lower para esophageal area and lesser omentum by the CT scan.

**Figure 4 medicina-59-01504-f004:**
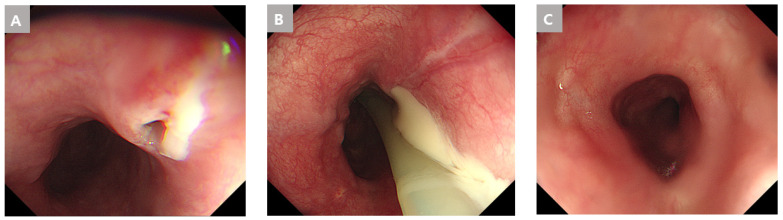
Endoscopic findings of the distal esophageal perforation with descending peritoneal abscess due to an esophageal fish bone foreign body in Case 2. (**A**) A mucosal small ulcer was observed with endoscopy. (**B**) Continuous drainage of whitish exudate at the perforation site was observed with endoscopy. (**C**) Disappearance of the mucosal ulcer with endoscopy.

## Data Availability

The data presented in this study are available on request from the corresponding author.

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
