# Peer review of "Two Cases of Severe Complications Due to an Esophageal Fish Bone Foreign Body"

_medicina, 2023, doi:10.3390/medicina59091504_

Round 1

Reviewer 1 Report

Dear authors,

Many thanks for allowing me to read your interesting article. Fishbones can be a nightmare and I congratulate the authors on safe outcomes of the two patients. I have few comments, I hope the authors find them acceptable. Once again thank you and well done !

Line 32-36

Esophageal fish bone FB diseases have a wide spectrum of clinical manifestations, 32 from minor diseases that can resolve spontaneously to severe fatal diseases that can lead 33 to death. Although rare, these diseases can lead to severe complications, including stric- 34 ture formation, esophageal perforation, tracheoesophageal fistula, and aortoesophageal 35 fistula. A significant delay in diagnosis can result in fatal complications.

Question: Please provide references

Line 36:

Question: For the benefit of a novice reader, please mention various techniques. Do the authors mean- Rigid endoscopy, flexible endoscopy??

Figure 1: Was the fish bone visualised on the CT scan?

Line 60:

Can you please confirm that this patient have oeosphageal resection for a foreign body ? If he has had resection, did the authors perform an end-to-end anastomosis?

Was the resected specimen sent for Histology?

Line 61:

Was the pus sent for microbiology, if so, what organisms grew?

Line 83:

Is this patient being followed up to see if he develops a stricture?

Line 85-86

This case is about distal esophageal perforation with descending peritoneal abscess in pa- 85 tients with esophageal fish bone foreign bodies

Should this be

This case is about distal esophageal perforation with descending peritoneal abscess in pa- 85 tients with esophageal fish bone foreign body

Line 88-89

Three days before, the patient got a fish bone stuck in their throat while eating grilled fish, and afterward, the patient had chest discomfort, abdomen pain, and dyspnea.

Can you consider

Three days prior, the patient  had a fish bone stuck impaction in their throat whilst eating grilled fish, and following this, the patient had chest discomfort, abdominal pain, and dyspnoea.

Line 91:

Dis the CT scan show a foreign body? Please mention if it was visible or not seen.

Line 136-137:

Fish bones are a very common cause of FB diseases in adults in Asia, more so than in 136 Western countries.

Please provide reference

Line 144-145:

As is known, the risk factors for the complications caused by fish bone FBs increase 144 with longer durations of impaction (>24 hours), bone type, and longer bone lengths 145 (>3cm). I

Please provide reference

English language and choice of words can be improved as suggested above 

Author Response

Thank you very much for your heartfelt review.
All items you pointed out have been corrected.
If you have any corrections after checking, please let me know.
Thank you again.

Reviewer 2 Report

in this study the authors presented two cases of foreign body disease resulting in esophageal abscess. 

The presentation of the cases are concise and crucial details are missing.

Case 1: Why they did not proceed in an endoscopy course first (which is a more conservative approach than VATS), before they lead the patient to the operating room? Please, describe with more detail the surgery performed. Where was the incision, what type of esophagectomy was performed, and how was this done, what were the exact intraoperative findings, are there any intraoperative photos? Provide normal ranges for all the laboratory results (e.g CRP). More details for the postoperative period would be very helpful. 

Case 2: Please, provide information about the clinical findings during clinical examination (this is missing in case 1, too). Line 99. I think the small mucosal opening is 4.0cm not 40cm.

In general, a more detailed presentation of the cases are needed.

Moderate editing is needed.

Author Response

Thank you very much for your heartfelt review.
All items you pointed out have been corrected.
If you have any corrections after checking, please let me know.
Thank you again.

In the first case, medistitinitis was determined by CT performed after visiting the emergency room at the hospital, and it was decided to perform drainage through surgery first. However, access was made through the NECK and esophagus during surgery, but abscess could not be drained. Therefore, endoscopy was performed later, and drainage was able to be performed through esophageal mucosa dissection.
The description of the case has been reinforced.
It has also been modified to reflect the points you pointed out.

Round 2

Reviewer 2 Report

The authors adressed all the issues that had been mentioned. The presentation of these interesting cases are sufficient. 

No major issues regarding English were found.